# Dynamic Localization of Paraspeckle Components under Osmotic Stress

**DOI:** 10.3390/ncrna10020023

**Published:** 2024-04-12

**Authors:** Aysegul Yucel-Polat, Danae Campos-Melo, Asieh Alikhah, Michael J. Strong

**Affiliations:** 1Molecular Medicine Group, Schulich School of Medicine & Dentistry, Robarts Research Institute, Western University, London, ON N6A 3K7, Canada; ayucelpo@uwo.ca (A.Y.-P.); asieh.alikhah@yahoo.com (A.A.); 2Department of Clinical Neurological Sciences, Schulich School of Medicine & Dentistry, Western University, London, ON N6A 3K7, Canada

**Keywords:** NEAT1_2, osmotic stress, paraspeckle proteins, cytoplasmic aggregates, membraneless organelles

## Abstract

Paraspeckles are nuclear condensates formed by NEAT1_2 lncRNA and different RNA-binding proteins. In general, these membraneless organelles function in the regulation of gene expression and translation and in miRNA processing, and in doing this, they regulate cellular homeostasis and mediate pro-survival in the cell. Despite evidence showing the importance of paraspeckles in the stress response, the dynamics of paraspeckles and their components under conditions of osmotic stress remain unknown. We exposed HEK293T cells to sorbitol and examined NEAT1_2 expression using real-time PCR. Localization and quantification of the main paraspeckle components, NEAT1_2, PSPC1, NONO, and SFPQ, in different cellular compartments was performed using smFISH and immunofluorescence. Our findings showed a significant decrease in total NEAT1_2 expression in cells after osmotic stress. Sorbitol shifted the subcellular localization of NEAT1_2, PSPC1, NONO, and SFPQ from the nucleus to the cytoplasm and decreased the number and size of NEAT1_2 foci in the nucleus. PSPC1 formed immunoreactive cytoplasmic fibrils under conditions of osmotic stress, which slowly disassembled under recovery. Our study deepens the paraspeckle dynamics in response to stress, suggesting a novel role for NEAT1_2 in the cytoplasm in osmotic stress and physiological conditions.

## 1. Introduction

Paraspeckles are nuclear membraneless organelles formed by micellization, a novel type of intracellular phase separation mechanism of biomolecular condensates [1,2,3]. These membraneless organelles are widespread in the nucleus of mammalian cells except for human embryonic stem cells (hESCs) [4]. Paraspeckles play crucial roles in gene expression by sequestering proteins from their gene regulatory functions and mRNAs from translation, thus controlling cellular homeostasis [5,6,7]. In addition, key paraspeckle components have been described to globally enhance pri-miRNA processing [8]. Paraspeckles are involved in development, stress response, differentiation, and diseases such as cancer and neurodegeneration [9,10,11].

Paraspeckles are formed by a specific long non-coding RNA (lncRNA), nuclear paraspeckle assembly transcript 1 (NEAT1), and over 40 different proteins through RNA–protein and protein–protein interactions. NEAT1 has two isoforms, NEAT1_1 (3.7 kb) and NEAT1_2 (23 kb), of which NEAT1_2 is essential for the recruitment of paraspeckle proteins (PSPs) [5,12]. The main paraspeckle proteins include non-POU domain-containing octamer-binding protein (NONO, also known as P54^nrb^), splicing factor proline/glutamine rich (SFPQ; also known as PTB-associated splicing factor (PSF)) and paraspeckle protein component 1 (PSPC1), all members of the drosophila behavior human splicing (DBHS) protein family and well-known paraspeckle components that form homo or heterodimers and modulate paraspeckle dynamics [13,14].

Evidence indicates that paraspeckles function as cytoprotective organelles that sense and respond to different stressors [9]. Several studies have shown that different stress conditions such as proteasome inhibition, mitochondrial stress, heat shock, and hypoxia elevate the number of paraspeckles and NEAT1_2 expression [5,15,16,17]. In osmotic stress, where alterations in the homeostasis of the cell volume trigger phase separation and formation of stress granules (SGs) and processing bodies (P-bodies), the effects on paraspeckles and their components remain unclear [18,19,20,21]. Alterations in the osmoregulation of the cell drive pathophysiological conditions observed in inflammation, cerebral edema, and aging [22]. Different studies using hyperosmotic stress have also observed that it leads to changes in the aggregation of neurodegeneration-associated proteins and their subcellular localization [23,24,25]. Therefore, studying the effect of osmotic stress on paraspeckle dynamics could shed light on understanding the underlying mechanisms of different diseases.

In this study, we investigated the effect of osmotic stress on the localization dynamics of the main paraspeckle components in HEK293T cells. We showed that under conditions of osmotic stress induced by sorbitol, total NEAT1_2 expression and the number of nuclear NEAT1_2 foci significantly decreased. We observed that osmotic stress changes the subcellular distribution of NEAT1_2; the main PSPs, PSCP1, NONO, and SFPQ; and the size of NEAT1_2 particles. This suggests a role for these paraspeckle components in the cytoplasm during stress.

## 2. Results

### 2.1. Osmotic Stress Decreases NEAT1_2 Expression

Previously, it has been reported that NEAT1_2 expression increases under various stress conditions [5,16,26]. To study if this also happens in osmotic stress, we incubated HEK293T cells with sorbitol and performed qPCR to detect total NEAT1_2 expression. Interestingly, we observed a significant reduction in NEAT1_2, which later returned to control levels after recovery (Figure 1A). We used smFISH and RNA probes against the middle segment of NEAT1_2 to confirm the qPCR results. After quantification we observed a significant decrease in total NEAT1_2 foci in cells under osmotic stress compared to the control condition (Figure 1B). As expected, HEK293T cells treated with the proteasome inhibitor MG132, known to induce NEAT1_2 expression and the number of paraspeckles [5,27], showed an increase in NEAT1_2 foci confirming the detection of NEAT1_2 (Appendix A).

### 2.2. Subcellular Localization of NEAT1_2 in HEK293T Cells

To observe details of the structure of NEAT1_2 and investigate changes under conditions of osmotic stress, we performed smFISH experiments using RNA probes against 5’ segments of the short and long variants of NEAT1 (NEAT_2-5’) and the middle segment of NEAT1_2 (NEAT1_2 middle) (Figure 2A). The long variant of NEAT1 (NEAT1_2) has been described as a nuclear lncRNA [28,29]. However, our results in HEK293T cells showed that NEAT1_2 was also localized in the cytoplasm under control conditions and osmotic stress (Figure 2B). To confirm that these were real cytoplasmic NEAT1_2 foci, we examined the conformation of NEAT1_2 described previously [28,30]. We used the structural features of nuclear NEAT1_2 foci in the control as a reference for cytoplasmic and nuclear NEAT1_2 foci in the control and sorbitol samples. Typical shell–core structures, in which the 5’ segment of NEAT1_2 localizes in the outer part of the foci and the middle segment in the core, were clearly observed under the control and osmotic stress conditions in the nucleus and cytoplasm. The signals obtained with the 5’ segment probes also presented the expected patchy pattern (Figure 2C) [30].

Using Z-stack imaging and 3D reconstructions, we detected NEAT1_2 foci of different sizes and conformations, including clusters (Figure 2D1,3) and single granules (Figure 2D4), in the nucleus and cytoplasm. Large single foci and clusters typically exhibited well-defined, regular, and organized structures in the nucleus and cytoplasm. Conversely, small NEAT1_2 foci showed higher structural variability and lacked the organized shell–core structure (Figure 2D4). We also observed some NEAT1_2 clusters partially residing within the nucleus and extending into the cytoplasm (Figure 2D2), which might suggest the transport of NEAT1_2 between the two subcellular compartments.

### 2.3. Number and Size of NEAT1_2 Foci Change under Osmotic Stress

Next, we studied how NEAT1_2 foci change under conditions of osmotic stress in the nucleus and cytoplasm. smFISH quantification showed a significant reduction in the number of NEAT1_2 foci within the nucleus and an increase in the cytoplasm under conditions of osmotic stress compared to the control condition (Figure 3A). Under normal conditions, the cytoplasmic NEAT1_2 foci percentage was 2.96%, while it increased to 19.67% under conditions of osmotic stress. During the recovery phase, the number of nuclear NEAT1_2 foci was restored to levels similar to control conditions, and no significant difference was observed in the number of cytoplasmic NEAT1_2 foci. Violin plots revealed that NEAT1_2 foci size within the nucleus was significantly decreased under conditions of osmotic stress, as well as the nuclear NEAT1_2 foci compared to their cytoplasmic counterparts (Figure 3B,C and Appendix A). There was no significant difference in cytoplasmic NEAT1_2 foci size between the control, osmotic stress, or recovery conditions, nor was there a difference between nuclear and cytoplasmic NEAT1_2 foci size in the control or recovery samples (Appendix A). These findings suggest that osmotic stress disrupts nuclear NEAT1_2 localization, shifting its localization to the cytoplasm.

### 2.4. Osmotic Stress Leads to Mislocalization of Paraspeckle Proteins

Previous studies have shown that some PSPs are recruited to the cytoplasm under stress conditions and disease [27,31,32]. Here, we observed that PSPC1, NONO, and SFPQ translocate from the nucleus to the cytoplasm under conditions of osmotic stress, colocalizing in protein condensates (Figure 4A and Appendix A). Interestingly, we observed that PSPC1 formed cytoplasmic fibrillar-shaped structures in a subset of cells under conditions of osmotic stress, which did not colocalize with NONO and SFPQ. These PSPC1 fibrils were fully formed after 4 h of sorbitol treatment and started to disassemble at 4 h of recovery (Figure 4B). After 8 h of recovery, when most of the PSPC1 had returned to the nucleus, some partially assembled PSPC1(+) fibrils were still observed in the cytoplasm (Figure 4A,B).

When we studied the dynamics of formation after osmotic stress, we observed that paraspeckles were present in low number in the nucleus of control HEK293T cells and had begun to disappear 30 min after the initiation of stress. After 4 h of sorbitol treatment, paraspeckles were barely visible in the nucleus (Figure 5A). Additionally, under conditions of osmotic stress, NEAT1_2 was observed in the cytoplasm alone or colocalizing with PSPC1, NONO, or SFPQ (Figure 5B). Altogether these observations suggest that HEK293T cells do not need paraspeckles to deal with osmotic stress and, in turn, maintain NEAT1_2 in the cytoplasm and translocate PSPs to potentially exert other functions.

### 2.5. Paraspeckle Components Partially Colocalize with SGs under Conditions of Osmotic Stress

Since RNA is recruited into SGs under certain conditions [33,34], we asked whether cytoplasmic NEAT1_2 colocalize with SGs. We observed that NEAT1_2 colocalizes with G3BP1(+) single SGs (Figure 6AI) and clusters (Figure 6AII) in the cytoplasm; however, the majority of cytoplasmic NEAT1_2 did not colocalize with these granules (Figure 6AIII,IV).

It has been reported that some PSPs, especially PSPC1, localize within SGs in the cytoplasm under oxidative stress [27]. Thus, we examined whether osmotic stress leads to the recruitment of PSPs to the SGs in HEK293T cells. We observed that, at 4 h of osmotic stress, PSPC1, NONO, and SFPQ colocalized with TIA1(+) SGs in a subgroup of cells. Similarly, we observed that PSPC1 and SFPQ colocalize with G3BP1 and TIA1(+) SGs, also in a small subpopulation of cells (Figure 6B). SGs and cytoplasmic colocalization with NEAT1_2 and PSPs were not detected in the control samples (Appendix A).

## 3. Materials and Methods

### 3.1. Cell Culture and Stress Treatment

HEK293T cells (ATCC) were cultured in Dulbecco’s modified Eagle’s media (DMEM; Gibco-Life Technologies, Grand Island, NY, USA) with 4.5 g/L glucose (25 mM), 10% fetal bovine serum (FBS; Gibco-Life Technologies), 1% penicillin–streptomycin (Pen-Strep; Gibco-Life Technologies) and 5 μg/mL plasmocin (InvivoGen, San Diego, CA, USA) at 37 °C with 5% CO_2_. For osmotic stress, HEK293T cells were exposed to 400 mM of sorbitol (Sigma, St. Louis, MO, USA) for 4 h. For recovery assays, cells were washed with PBS once and maintained in DMEM for 4 h. To induce proteasome inhibition, cells were incubated with 10 µM of MG132 for 4 h. For fluorescence in situ hybridization (FISH) and immunofluorescence, cells were seeded on coverslips coated with attachment factor protein (Gibco-Life Technologies). Cells on coverslips were fixed with 4% paraformaldehyde (PFA)-PBS (Electron Microscopy Sciences, Hatfield, PA, USA, 32%) solution for 10 min and washed with PBS.

### 3.2. Quantitative Real-Time PCR (qPCR)

Total RNA was isolated from HEK293T cells using TRIzol™ reagent (Life Technologies, Carlsbad, CA, USA). RNA concentration and purity were checked using spectrophotometry. The needle shearing method was applied to samples in TRIzol during the RNA isolation step to increase the extractability of NEAT1_2 [35]. Then, cDNA was synthesized using SuperScript IV VILO Master Mix with ezDNease Enzyme (Invitrogen, Carlsbad, CA, USA) according to the manufacturer’s instructions. Real-time PCR experiments were performed on the ViiA 7 Real-Time PCR System using TaqMan Fast Advanced Master Mix (Applied Biosystems, Waltham, MA, USA) and TaqMan Assay for NEAT1_2 (Hs01008264_s1). GAPDH expression (Taqman Assay; Hs02758991_g1) was used as a normalizer.

### 3.3. Single-Molecule Fluorescence In Situ Hybridization (smFISH)

After fixation, cells were stored in 70% ethanol at 4 °C overnight for cell permeabilization. Commercially available probe sets against the NEAT1 5’ segment and middle segment (Stellaris FISH probes, NEAT1 5’ segment with Quasar 670 dye, and NEAT1 middle segment with Quasar 570 dye, Biosearch Technologies, Teddington, UK) and wash and hybridization buffers (Biosearch Technologies) were used according to the manufacturer’s instructions. Hybridization was performed at 37 °C for 5 h. Coverslips were mounted with Prolong Diamond (Invitrogen, Waltham, MA, USA). Slides were stored at −20 °C.

### 3.4. Immunofluorescence

Fixed cells were permeabilized using 0.2% Triton X-100 in PBS for 10 min and incubated with 50 mM ammonium chloride in PBS to quench PFA for 30 min. Cells were incubated with 4% bovine serum albumin (BSA) in PBS for 1 h at room temperature (RT) with primary antibody: mouse anti-PSPC1 (1:100, Sigma, St. Louis, MO, USA, Cat. No. SAB4200503), rabbit anti-PSPC1 (1:100, Abcam, Cambridge, UK, Cat. No. ab104238), mouse anti-NONO (1:100, BD Transduction, Franklin Lakes, NJ, USA, Cat. No. 611279), rabbit anti-SFPQ (1:100, Proteintech, Rosemont, IL, USA, Cat. No. 15585-1-AP), mouse anti-G3BP1 (1:300, Proteintech, Cat. No. 66486-1-Ig), or goat anti-TIA1 (1:100, Santa Cruz, CA, USA, Cat. No. sc-1751). Following washing, Alexa Fluor 488, Alexa Fluor 555, or Alexa Fluor 633 secondary antibodies were applied to coverslips for 1 h at RT. Nuclei were stained with Hoechst (2 µg/mL).

### 3.5. Confocal Microscopy and Image Analysis

Samples were imaged using a Leica TCS SP8 confocal microscope and LAS X software (Version 3.5.5.19976) using the LIGHTNING module. Z-stack series images were used to generate 3D image reconstructions using the Leica LAS X 3D Viewer. One hundred cells per slide across three slides of control, sorbitol, and recovery samples were used for quantification and size measurement of NEAT1_2 foci. Manual analysis was conducted using the “analyze particles” function in the FIJI/ImageJ software (Image J 2.9.0/1.54f) (National Institutes of Health).

### 3.6. Statistical Analysis

Graphs and statistical analyses were performed using GraphPad Prism software (Version 10.2.0 (392)). Students *t*-tests and one-way ANOVA with Tukey post hoc tests were used to compare two or multiple groups. Results were considered significant at *p* < 0.05.

## 4. Discussion

In this study, we investigated the effect of osmotic stress in the dynamics of subcellular localization of paraspeckle components. We observed a significant decrease in total NEAT1_2 expression and the size of nuclear NEAT1_2 foci under osmotic stress in HEK293T cells. Interestingly, we showed that NEAT1_2 localizes in the cytoplasm, in addition to its well-defined nuclear localization. Our results suggest that osmotic stress leads to the translocation of paraspeckle components, NEAT1_2, PSPC1, NONO, and SFPQ from the nucleus to the cytoplasm, and that PSPC1, in particular, forms fibrillar structures in the cytosol under conditions of osmotic stress.

Osmotic stress, unlike other types of stress conditions, induces physical alterations in cells, such as reduced cell volume and shrinkage of the nucleus due to water loss. These alterations cause molecular crowding, increase molecule concentration, and can trigger phase separation with membraneless organelle formation [18,36]. It has been shown that PSPC1, NONO, and SFPQ can form heterodimers and complexes [13,37]. SFPQ and NONO have nuclear export signals in addition to their nuclear retention signals, and NONO can be continuously exported from the nucleus to the cytoplasm with an exportin 1-independent pathway [38]. Additionally, the phosphorylation of SFPQ leads its shuttling to the cytoplasm [39]. PSPs translocate to the cytoplasm and form aggregates in the stress response, neurodegenerative diseases, and cancer [13,31,40,41]. This evidence is in line with our results in which we observed a nucleocytoplasmic shift of paraspeckle proteins and the formation of cytosolic condensates under osmotic stress. Although the transport mechanism of PSPs has not been fully identified yet, the evidence suggests that an active transport mechanism leads to their translocation from the nucleus to the cytoplasm.

Osmotic stress also induces the formation of some stable fibrillar cytoplasmic aggregates of different proteins like TAU and polyglutamine (polyQ)-repeat–containing proteins [42,43]. The observation of the delayed resolution of PSPC1(+) fibrils following removal of the osmotic stress is unique for this group of paraspeckle proteins, since NONO and SFPQ formed typical condensates of irregular shape that rapidly resolved under recovery conditions.

In addition, we observed the colocalization of PSPs with TIA1 and G3BP (+) SGs in a small percentage of cells in the osmotic stress samples. This observation is consistent with findings that G3BP1 colocalizes with NONO and SFPQ in mouse retinal ganglion cells in a stress-independent but differentiation-related manner [44]. Cytoplasmic aggregates containing SFPQ and TIA1 have also been observed in an in vivo Alzheimer’s disease model [45]. However, our findings revealed a different colocalization pattern of PSPs with SGs under osmotic stress from what has been reported in the literature for oxidative stress. For example, broad colocalization of PSPC1 with TIAR, an SG marker, it has been observed, while no SFPQ colocalization was found in SGs under oxidative stress [27]. This distinctive pattern of concurrent localization suggests a unique response of these proteins to osmotic stress.

Few studies have suggested that NEAT1_2 could have function(s) in the cytoplasm beyond nuclear paraspeckle assembly. A mutation in *ASLXL1*, a gene that encodes for a histone modifier, has been shown to induce an increase in the cytoplasmic localization of NEAT1_2, NONO, and SFPQ in mouse hematopoietic stem and progenitor cells [46]. In breast cancer, the cytoplasmic localization of NEAT1_1 has been linked to a putative role in glycolysis [47]. Our observation of NEAT1_2 in the cytoplasm, with or without PSPs even under control conditions, and the absence of visible alterations in the nuclear membrane integrity observed with confocal microscopy, suggest that there is active transport of NEAT1_2 from the nucleus to the cytoplasm. Maintaining certain levels of this NEAT1_2 in the cytosol might be necessary for the cell to cope with osmotic stress because this shuttling occurs even when sorbitol causes nuclear pore constriction, thus reducing nuclear transport [48]. Studying the transport of NEAT1_2 and its function in the cytoplasm in normal and pathological conditions could provide a broader view of the physiological role of this lncRNA.

Many studies have demonstrated elevated paraspeckle numbers and NEAT1_2 expression in various stress conditions. The consensus is that this increase facilitates cellular homeostasis and cell survival. However, in our study we observed a significant decrease in NEAT1_2 expression and NEAT1_2 foci in the nucleus under osmotic stress [5,16,17]. A similar reduction was reported before in the presence of chemicals that induced DNA single-strand breaks (SSBs) [49]. NEAT1_2 levels depend on its transcriptional activity and stability. It has been shown that proteins such as breast cancer susceptibility gene 1 (BRCA1) [50] and E2F transcription factor 1 (E2F1) [51] reduce NEAT1_2 transcription. On the other side, SFPQ and NONO have essential roles in NEAT1_2 stabilization [12,52,53]. Other proteins that stabilize NEAT1_2 include serine/arginine-rich splicing factor 1 (SRSF1), polymerase 1 and transcript release factor (PTRF/Cavin-1), arsenic resistance protein 2 (ARS2), and AU-binding factor 1 (AUF1) [54,55,56,57]. The reduction we observed in nuclear NEAT1_2 suggests that paraspeckles are not needed for the cell to cope with osmotic stress. Because of this, repression of the transcription and/or destabilization of NEAT1_2 might be necessary. Determining which processes and proteins are critical for NEAT1_2 changes could be an interesting goal for future studies.

The present findings indicate cytoplasmic localization of NEAT1_2 and the nucleocytoplasmic shuttling and aggregation of paraspeckle components under conditions of osmotic stress. Understanding the cytoplasmic dynamics and functions of paraspeckle proteins and NEAT1_2 in stress conditions may contribute to revealing their role in different diseases.

## Figures and Tables

**Figure 1 ncrna-10-00023-f001:**
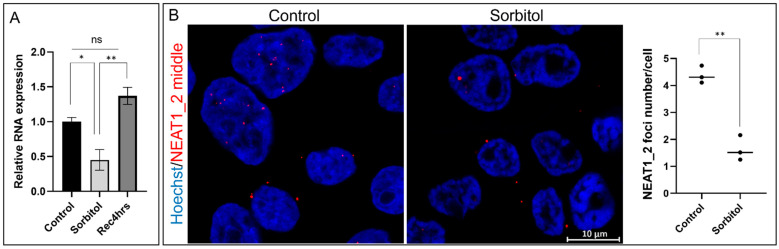
NEAT1_2 expression decreases under the conditions of osmotic stress in HEK293T cells. (**A**) qPCR shows that total NEAT1_2 expression was significantly reduced under conditions of osmotic stress (sorbitol, 4 h) and returned to control levels after 4 h of recovery (Rec, 4 h) (ΔCt values were used for statistical analyses, one-way ANOVA, * *p* < 0.05, ** *p* < 0.01, *n* = 5, error bars indicate SEM). (**B**) Single-molecule FISH (smFISH) using Stellaris RNA probes against the NEAT1_2 middle segment shows that the number of NEAT1_2 foci in the cells decreased under conditions of osmotic stress (*t*-test, ** *p* < 0.01).

**Figure 2 ncrna-10-00023-f002:**
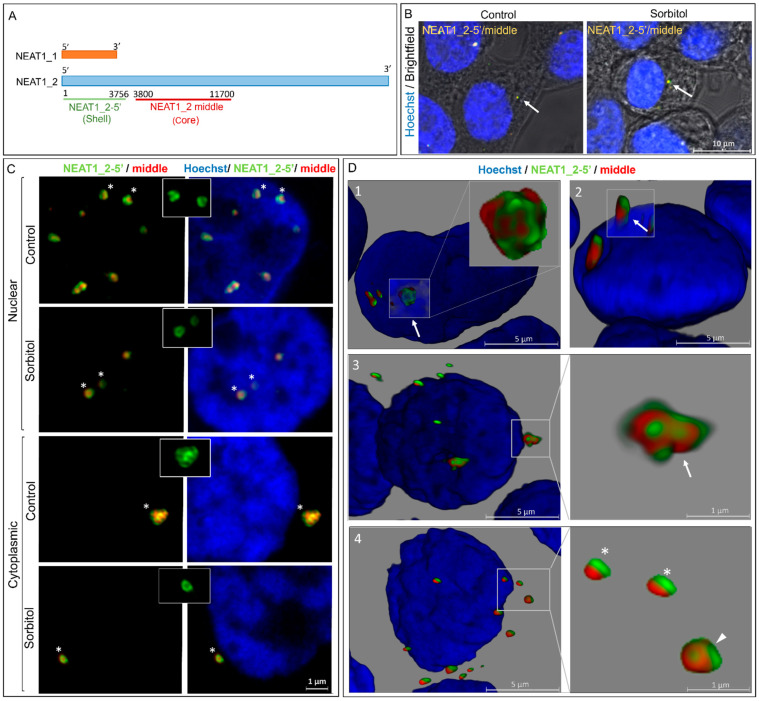
NEAT1_2 localizes in the cytoplasm of HEK293T cells forming typical shell–core structures. (**A**) Schematic representations of the positions of the smFISH probes used to identify distinct segments of NEAT1. (**B**) NEAT1_2 foci are observed in the cytoplasm at baseline and under conditions of osmotic stress (arrows). (**C**) The shell–core structure of NEAT1_2 (5′ segment in the shell and middle segment in the core) was detected in both cytoplasmic and nuclear NEAT1_2 foci in the control and sorbitol samples. Insets show the patchy pattern of the 5′ segment of NEAT1_2 foci. (asterisk used to demonstrate the granules in the insets) (**D**) 3D reconstruction shows NEAT1_2 foci of varying sizes and conformations in both the nuclear and cytosolic compartments. This includes clusters (arrows in 1 and 3) and single granules (arrowhead in 4) with shell–core structures in the nucleus (1) and cytoplasm (3 and 4) in the sorbitol samples. Small granules that, which lack the shell–core structure and showed the 5′ and middle segments of NEAT1_2 side by side, were also observed in the nucleus and cytosol under conditions of osmotic stress (asterisk in 4). Some NEAT1_2 clusters were observed partially residing within the nucleus and extending into the cytoplasm (arrow in 2).

**Figure 3 ncrna-10-00023-f003:**
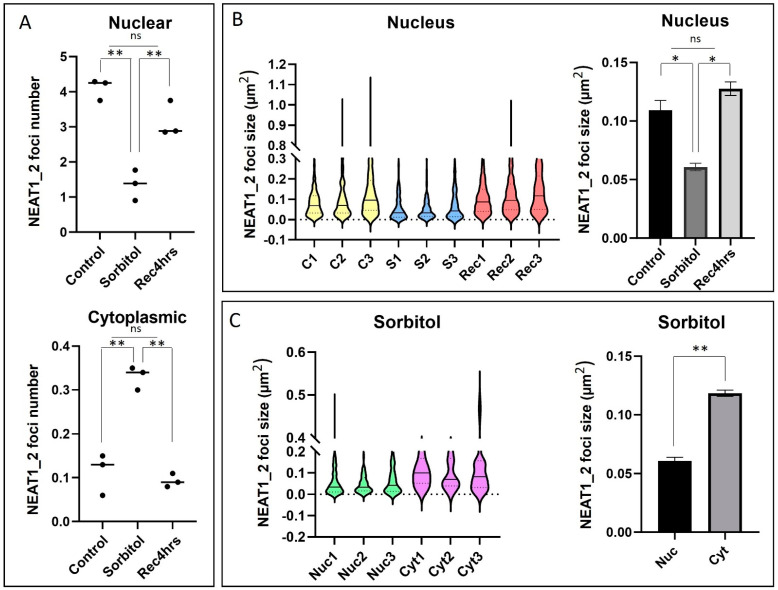
The number and size of NEAT1_2 foci change in HEK293T cells under osmotic stress. (**A**) Quantification of smFISH experiments showed that the number of nuclear NEAT1_2 foci decreased under conditions of osmotic stress, while the cytoplasmic foci increased (100 cells per n, n = 3, one-way ANOVA, ** *p* < 0.01). (**B**) Violin plots show the distribution of NEAT1_2 foci size under different conditions. A decrease in NEAT1_2 foci size in the nucleus was observed under osmotic stress compared to the control and 4 h recovery (C: control; S: 4 h sorbitol; Rec: recovery-4 h; 100 cells per n, n = 3, One-way ANOVA, * *p* < 0.05, error bars indicate SEM). (**C**) Under conditions of osmotic stress, cytoplasmic NEAT1_2 foci were significantly larger than nuclear NEAT1_2 foci (100 cells per n, n = 3, *t*-test, ** *p* < 0.01, error bars indicate SEM).

**Figure 4 ncrna-10-00023-f004:**
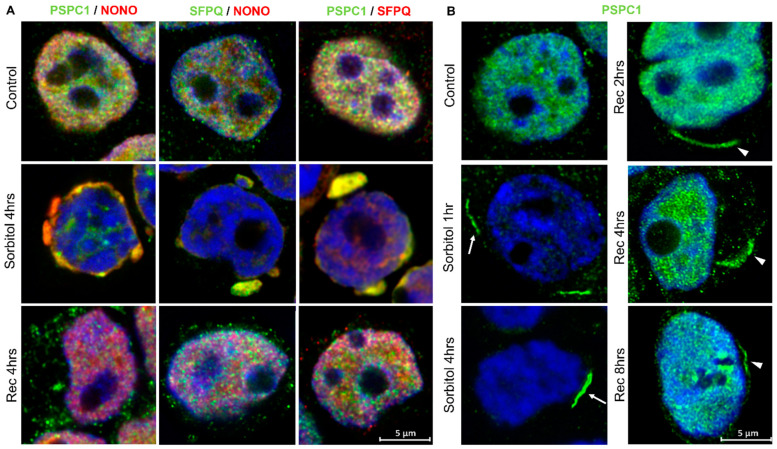
Paraspeckle proteins colocalize in the cytoplasm of HEK293T cells after osmotic stress. (**A**) Under conditions of osmotic stress, PSPC1, NONO, and SFPQ redistributed from the nucleus to the cytoplasm where they colocalized. Pictures show the merging of the PSP staining. (**B**) After osmotic stress, PSPC1 immunoreactive fibrils were formed in the cytoplasm (arrows). After 8 h of recovery, cytoplasmic PSPC1(+) fibrils were observed partially assembled (arrowhead), and SFPQ and NONO returned to the nucleus. Nuclei were stained using Hoechst (blue).

**Figure 5 ncrna-10-00023-f005:**
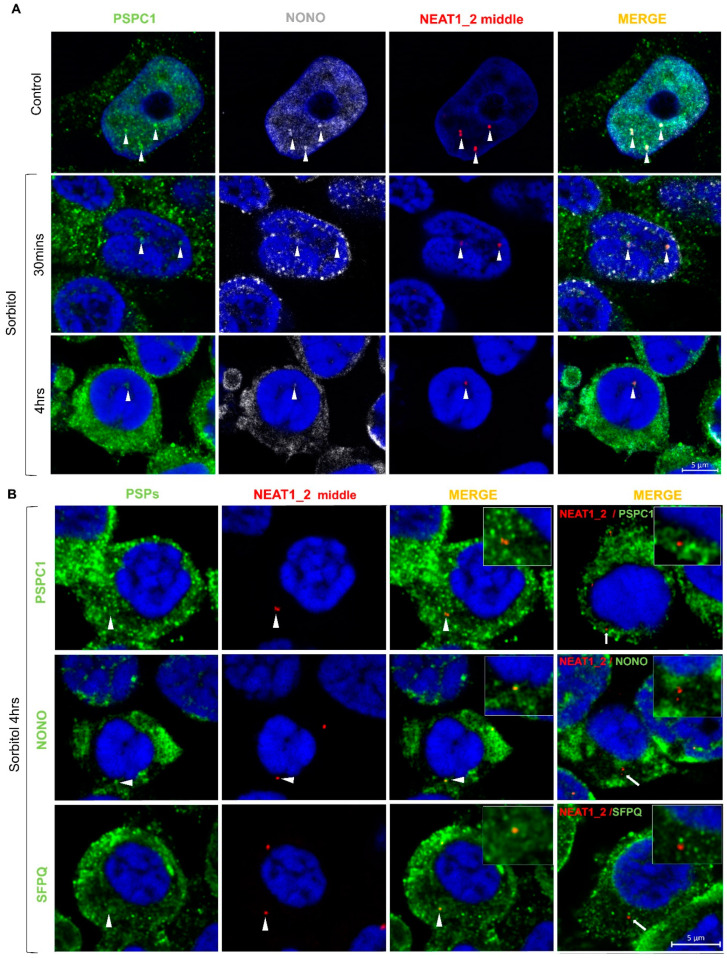
Colocalization of NEAT1_2 with PSPs in the nucleus and cytoplasm of HEK293T cells. (**A**) smFISH/IF experiments showed that PSPC1, NONO, and NEAT1_2 form paraspeckles in the nucleus as expected in the baseline condition. After 30 min of incubation with sorbitol, PSPC1 and NONO redistributed to the cytosol, and paraspeckles began to disappear. After 4 h of sorbitol treatment, paraspeckles were barely visible in the nucleus. (**B**) NEAT1_2 localizes in small granules in the cytosol with (arrowheads) or without PSPC1, NONO, and SFPQ (arrows) under osmotic stress.

**Figure 6 ncrna-10-00023-f006:**
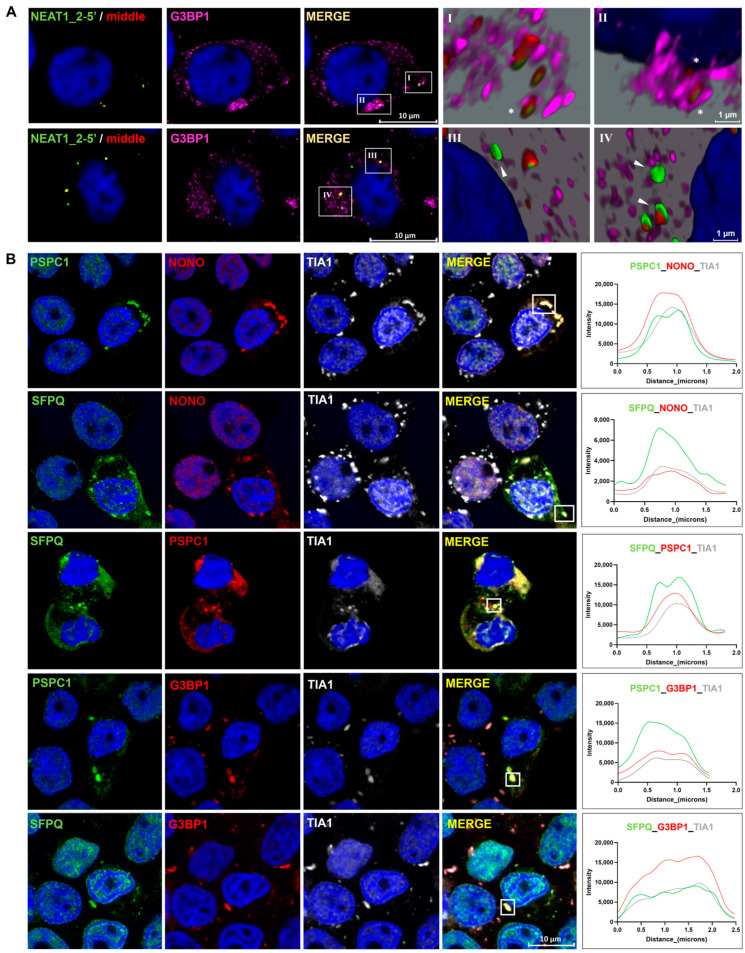
Paraspeckle components colocalize with stress granules in the cytoplasm under conditions of osmotic stress. (**A**) Colocalization images showed that cytoplasmic NEAT1_2 rarely colocalizes in G3BP1(+) SGs (I and II, asterisk). No extensive colocalization was observed (III and IV, arrowhead). (**B**) PSPC1, NONO, and SFPQ were present together in SGs after osmotic stress but only in a small subpopulation of the cells. Line profiles of the fluorescence intensities on the right show the colocalization of PSPs with TIA-1 and G3BP1 in areas in the squares.

## Data Availability

Dataset available on request from the authors.

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
