# Peer review of "Dynamic Localization of Paraspeckle Components under Osmotic Stress"

_ncrna, 2024, doi:10.3390/ncrna10020023_

Round 1

Reviewer 1 Report

Comments and Suggestions for Authors

The manuscript “Dynamic localization of paraspeckle components under osmotic stress” by Yucel-Polat and co-authors examined the dynamic of paraspeckles and their components under condition of osmotic stress. For that, authors performed smFISH with probes to the paraspeckle resident long non-coding RNAs and immunofluorescence for the main paraspeckle components as well as real-time PCR. The data on decrease in total nuclear paraspeckle assembly transcript (NEAT1_2) expression in HEK293T cells after stress conditions is presented. The authors clearly demonstrate the appearance of cytoplasmic NEAT1_2 foci with a typical shell-core structure and their colocalisation with PSPs proteins.

The manuscript is logically written and the data are really convincing with relevant control experiments. The manuscript is accompanied by high-quality micrographs and 3D reconstructions. Statistical analysis is also provided. In order to present the data in a clearer way, the following questions should be addressed.

Major comments:

Q1. Since paraspeckle proteins partially colocalise with SGs under osmotic stress, it would be beneficial to check whether some cytoplasmic NEAT1_2 foci colocalise with SGs under the same conditions.

Minor comments:

Line 16 “Our findings showed a significant decrease in total NEAT_2 expression in cells after osmotic stress.” – Should it be “NEAT1_2”?

Author Response

Major comments:

Since paraspeckle proteins partially colocalise with SGs under osmotic stress, it would be beneficial to check whether some cytoplasmic NEAT1_2 foci colocalise with SGs under the same conditions.

Following the reviewer’s suggestion, we performed FISH/IF and confocal microscopy under control and sorbitol conditions and observed that some NEAT1_2 colocalizes with SGs (single granules and clusters) under osmotic stress. We added these photos to Figure 6 and controls of the experiment to Supplementary Figure 4 in the new version of the manuscript.

Minor comments:

Line 16 “Our findings showed a significant decrease in total NEAT_2 expression in cells after osmotic stress.” – Should it be “NEAT1_2”?

Thank you for the comment. The typo was corrected in the manuscript.

Reviewer 2 Report

Comments and Suggestions for Authors

The study reports that under osmotic stress, paraspeckle components NEAT1 lncRNA and other associated proteins diminish, and that they are relocated from nucleus to cytosol, showing a dynamic nature. Below are some comments:

I think the most interesting aspect is that the new sites in cytosol align with stress granules. That being said, this phenomenon of NONO and SFPQ relocating to SG was previously reported. I think they can highlight that osmotic stress also causes the similar translocation, which still seems a novel finding.

My question then is what does osmotic stress do, and why is it important to know the effects of Sorbitol? Does it do similar effects as Arsonite, a well known SG-inducer?

The paper starts by saying that the “role” of these paraspeckle is unknown, but there is no description of these roles, only to highlight the trasnslocation itself.

Is the translocation from nucleus to cytosol a specific regulation (via export pathway) or could it be due to non-specific weakening of nuclear membrane due to Sorbitol treatment? The latter is unlikely but I still wonder.

Fig2C is a bit confusing; the boxed green signals on the left is not shown in the image on the right with DAPI. What does this figure try to emphasize?

Please provide the methods for 3D image reconstruction.

Author Response

  1. I think the most interesting aspect is that the new sites in cytosol align with stress granules. That being said, this phenomenon of NONO and SFPQ relocating to SG was previously reported. I think they can highlight that osmotic stress also causes the similar translocation, which still seems a novel finding.

We appreciate the reviewer's feedback. We highlighted the consistency of our findings with the literature in the discussion, lines 289-292. In the new version of the article, we emphasized the novelty of our findings related to sorbitol and added the comparison with oxidative stress in lines 293-298.

  1. My question then is what does osmotic stress do, and why is it important to know the effects of Sorbitol? Does it do similar effects as Arsonite, a well known SG-inducer?

The distinct mechanism of osmotic stress compared to other stress conditions was described in the discussion section between lines 268 and 271. Now, we also included the importance of studying osmotic stress in the introduction section between lines 55-61.

  1. The paper starts by saying that the “role” of these paraspeckle is unknown, but there is no description of these roles, only to highlight the translocation itself.

The role of paraspeckles was explained in the introduction section between lines 33 and 38 and 49-50. In this new version of the article, we extended this part in the abstract section, lines 11-14.

  1. Is the translocation from nucleus to cytosol a specific regulation (via export pathway) or could it be due to non-specific weakening of nuclear membrane due to Sorbitol treatment? The latter is unlikely but I still wonder.

We appreciate the reviewer's comment. We have added information regarding paraspeckle proteins shuttling to the cytosol to the discussion of the article in lines 272-276. Also, we have not detected visible alterations in the integrity of the nuclear membrane under sorbitol. Currently, we are working on understanding the mechanism(s) for NEAT1_2 translocation to the cytosol. The evidence we have so far suggests active transport of paraspeckle components, even under sorbitol where nuclear pore constriction has been described. We added this information to the discussion section, between lines 304-310.

  1. Fig 2C is a bit confusing; the boxed green signals on the left is not shown in the image on the right with DAPI. What does this figure try to emphasize?

Thank you for this helpful observation. In this new version, we included insets of the green channel (5' segment probe) between the two columns of Fig2C to show the expected patchy pattern in the shell of NEAT1_2 foci.

  1. Please provide the methods for 3D image reconstruction.

The method for 3D image reconstruction was included in the materials and methods section between lines 113-114.

Reviewer 3 Report

Comments and Suggestions for Authors

The study by Yucel-Polat et al titled ‘Dynamic localization of paraspeckle components under osmotic stress’ characterizes the subcellular localization of the lncRNA NEAT1 and a few associated paraspeckle-related proteins (namely NONO, SFPQ, and PSPC1) in HEK-293 cells under osmotic stress. The authors observed a cytoplasmic localization of these paraspeckle components following 4 hours of treatment with 400mM sorbitol. The study expands the authors’ prior work on RNA metabolism, stress granules, and their association with disease. The experiments presented here are well-conceived. The study however is a descriptive study; there are no functional experiments that would lead to a mechanistic description of the events, which limits the impact of the work.

Major comments

As mentioned above, the study lacks a description of a mechanism for the nucleocytoplasmic translocation of paraspeckle components under osmotic stress, or experiments that would detail what the functional consequences of the cytoplasmic localization of these components are. Given that this would require a new line of experiments, including genetic and pharmacological interventions, this reviewer understands that such an undertaking may be out of the scope of this manuscript.

The experiments in the manuscript are well conceived and performed. The only concern that this reviewer has is specificity. There are three main comments here:

1)      How specific is the observed effect? While the reversibility of the effect does indicate that the translocation is specifically due to the sorbitol, there was no mention in the materials and methods on whether glucose was present in the media (since there are different types of DMEM, the levels of glucose should be clarified). Sorbitol is an inert sugar only in the presence of glucose.

2)      Does the observed translocation take place in other cell lines? Is it possible that the effect is a peculiarity of the HEK cells? The experiments in Figures 1 and 2 could be repeated with another cell line to ensure that the effect is not specific. Cos-7 cells are a great cell line for imaging, but any cell line available in the lab should work.

3)      Is the effect specific to paraspeckles? What about other nuclear bodies, e.g speckles? The observed translocation may or may not be specific to paraspeckles, and this would not truly affect the observations in the paper, but does MALAT-1 translocate under osmotic stress? This would provide some insight as to whether the observed effect is specific.

Minor comments

In the materials and methods, there is mention of HeLa cells, but there is no result described with HeLa cells. (Still, consider using them for point #2 in major comments above?)

Author Response

we have appended the response as an attachment (pdf) since it contains a figure

Round 2

Reviewer 2 Report

Comments and Suggestions for Authors

There are improvements in the revised version.

I still think the novelty is moderate, but the findings should help the field.

Reviewer 3 Report

Comments and Suggestions for Authors

The authors addressed the concerns raised following the first manuscript review. This reviewer is looking forward to their follow-up work that will hopefully shed light on the mechanism of paraspeckle translocation to the cytoplasm upon osmotic stress.